# Analysis of Road Users' Risk Behaviors in Different Travel Modes: The Bangkok Metropolitan Region, Thailand

Pawinee Iamtrakul [1,*] , Sararad Chayphong [1], Emese Makó [2,*] and Souvathone Phetoudom [2]

1 Center of Excellence in Urban Mobility Research and Innovation, Faculty of Architecture and Planning, Thammasat University, Pathumthani 12120, Thailand
2 Department of Transport Infrastructure and Water Resources Engineering, Széchenyi István University, 9026 Győr, Hungary
* Correspondence: pawinee@ap.tu.ac.th (P.I.); makoe@sze.hu (E.M.)

**Abstract:** This study explores road users' behaviors and accident analyses on different travel modes in the Bangkok Metropolitan Region (BMR). The questionnaire survey was adopted and designed based on contributing factors related to risk behavior perceptions in different travel modes. A total of 3000 participants submitted questionnaires that provided data for a multiple regression model analysis. The results indicated that different travel modes have different risk behavior perceptions. Road users of vulnerable modes of travel, such as walking and cycling, were more aware of risky behaviors than users of others. Risky, violent behavior may occur due to fatigue, driving while taking drugs, or aggressive tactics where the driver may lose control and cause an accident. However, driver negligence, especially violating traffic rules, may sometimes cause risky behavior. The results show that age, gender, education level, income level, marital status, driving experience, accident experience, and attitude toward road safety affect risk behavior perception. In conclusion, the differentiation of vehicles plays a vital role as a critical issue that should be understood for effectively mitigating risks in different travel patterns.

**Keywords:** accident analysis; mode choice; risk behavior; road safety; sustainable transport

## 1. Introduction

Travel is at the core of driving citizens' daily lives, including in urban developments, to improve quality of life and sustainable development. Mobility via transportation systems can connect to meeting personal needs between the origins and destinations of people within the city, such as access needs and social networks [1]. A transportation system should provide various travel options, including personal vehicles, public transportation, paratransit, and active transportation. Each travel mode reflects a choice people make for a variety of activities in cities. However, the challenge of traveling with different modes of transportation is inevitably related to road safety [2], especially when traveling on roads with a mixture of vehicles in the traffic. A transportation system encompasses a complex physical scale and a broad spectrum of activities within the transport route environment. Such complexity may, if improperly planned, lead to traffic conflicts that induce road accidents. Road accidents bring loss of life and property, directly affecting the victims, their families and their friends, and cause other issues, such as decreases in economic value and psychological suffering [3–6]. There are chances of accidents in each form of travel, especially by private vehicles (both motorcycles and private cars), while public transport has lower accident statistics.

However, road safety is an important issue that many countries pay attention to due to fatalities statistics. Road accidents, including in Thailand, are among the top causes of death in the world's population [7]. Therefore, many countries are struggling to find measures to prevent accidents regarding the behaviors of drivers, vehicles, roads, and related environments. The target is to reduce the number of fatalities from accidents by

half, as proposed by the Decade of Action for Road Safety 2021–2030 [8]. Although death rates have decreased, the reduction has not been enough [9]. Therefore, determining the root cause of road accidents is a fundamental goal.

Several studies have attempted to associate factors with road safety issues. The association of different risk behaviors while traveling in different modes of vehicles is another crucial issue, due to different collision protection systems and driving travel modes. The associated severity level may vary from one case to another. For a few instances, Scholes et al. [10] studied the fatality rates associated with driving and cycling. Their findings indicated that driving-related fatality rates were higher than were the rates for cycling. Aldred et al. [11] studied how the mode of travel affects risks posed to other road users, and they found that motorcycles pose a substantially higher per-km risk to others than cars.

In contrast, the study of Teschke et al. [12] found that the risks of driving, walking, and bicycling were similar to the respective risks of undertaking such activities in British Columbia. Each study provided findings on mortality and injury rates, including travel risks according to different travel patterns. Factors contributing to the risk of traveling in a different mode may arise from individual aspects such as age [13,14], gender [15], and attitude and perception of an individual road user. Furthermore, it may be related to facilitating modes of transport that may induce different levels of risky behavior. Vulnerability depends not only on road users (children, older adults) but also on fragile modes of travel. Many studies have examined the vulnerability of vehicles (bicycles and motorcycles) that do not protect the rider (non-shielded modes), creating more severe impacts and injuries. The above factors point to differences in the severity of accidents among travel modes. Therefore, the differentiation of vehicles is another critical issue that should be understood. This study focuses on the factors that affect road users' risk behaviors among different travel modes in the Bangkok Metropolitan Region (BMR).

## 2. Literature Review

### 2.1. Impact of Driver Behavior on Road Safety

Casualty resulting from road traffic accidents is the leading cause of death for road users in all age groups; more than half are vulnerable road users [16]. Accidents concern many sectors and remain a serious public health problem in many countries [7,16]. Although fatalities due to road traffic accidents have decreased compared to the past year, the overall number of incidents remains higher than the set target. Therefore, many countries still have policies or measures to reduce road accidents. The United Nations General Assembly has set an ambitious target of halving the global number of casualties from road traffic accidents by 2030 [7]. Drivers' risk behaviors are a crucial cause of accidents, and past studies suggest that this factor contributes to road traffic accidents among all road users [17–19]. Several factors drive road user behavior, such as demographic profiles (gender, age, etc.) [15,20], socioeconomic characteristics (marital status, income level, educational level) [21,22], road environment [23], risk perception [24], personality (attitude) [25], external or internal norm [26], and driving experience (accident experience and driving experience). However, the factors that drive human behavior are complex combinations; thus, many researchers have attempted to delve deeper into the behaviors related to road traffic accidents.

### 2.2. Risk Behaviors of Road Users in Different Travel Modes

The main factors that cause road accidents include road users, vehicles, roadways, and environmental factors [27]. Road users engage in many risky behaviors while traveling on the road, varying from intentional to unintentional. Past studies have investigated many risky road behaviors and divided them into several categories (e.g., violations, aggressive driving, personal errors, and distractions.) [28]. Aggressive driving corresponds with deliberate behaviors that endanger and may increase the risk of a collision, such as honking at other road users and overtaking [29,30]. Violation behavior is the deliberate infringement

of some regulated or socially accepted code of behavior, such as illegal crossing, not using a seatbelt or helmet, drunk driving, and running a red light [31,32]. In terms of personal errors, it represents the failure of planned actions to achieve their desired outcome without intervention, such as stopping suddenly and not switching on lights [31]. These are the concerns among several contributors to road traffic accidents. Finally, distractions shift or divert attention from driving, reducing concentration or focus when driving, such as listening to music, making phone calls, smoking, and eating [30,33]. Many studies have attempted to explain road users' risk behaviors, from various perspectives. Economists use expected utility theory to explain and predict real-world risk behavior [34] in psychological models that relate to the theory of planned behavior or theories of reasoned action [35–37]. However, the study of risk behavior through different models has limitations, suggesting that risk studies involve many factors that vary across the context of the environment and society.

Road users travel on the road or in different transportation network systems divided into many types (e.g., private cars, motorcycles, bicycles, pedestrians, public buses, rapid mass transit, and taxis). Every travel system must consider travel safety as different vehicles travel together, particularly some modes vulnerable to road users, such as pedestrians and cyclists. However, the system still plays a part in the movement, especially when walking or cycling across the road, which cuts off the main traffic on the road. Notably, the differences in vehicles do not appear only in physical characteristics and appearance (such as size of vehicle, protected by an outside shield, etc.), but other factors are also involved in choosing different modes of traveling, especially the fundamental factors related to demographic profile, socioeconomic characteristics, and personal characteristics. In some cases, there are restrictions on economic status, thereby limiting the use of traveling modes and limiting access to education or information on safe driving. Therefore, understanding the differentiation of vehicles is essential, as different vehicles have different vulnerabilities.

## 3. Methodology

### 3.1. Study Area

Notably, over 90 percent of the world's fatalities in road traffic accidents occurred in low- and middle-income countries (the most recent (2022) categories range from USD 1045 or less for low-income countries to an upper range of USD 4096–USD 12,695 for upper-middle-income countries) [7,38]. Thailand is one of the middle-income countries that have serious road traffic accidents. Each day Thailand behaves in ways that increase the risk of a road accident [39,40]. For example, more than 80 percent are accidents among motorcycle users. Concerning road traffic fatalities data (2017–2021) for all travel modes, data showed 96,230 deaths (an average of 15,000–19,000 deaths per year). For pedestrian accidents, road traffic fatalities data showed an average of 513 deaths per year [40,41]. This study selects the BMR as a case study, with Bangkok as the city center and the surrounding area of five provinces: (1) Nakhon Pathom, (2) Pathum Thani, (3) Nonthaburi, (4) Samut Prakan, and (5) Samut Sakhon (Figure 1). These urban areas result in creating various travel modes and have one of the highest accident statistics in the country [42]. Accidents have resulted from many causes, including human error, vehicles, roads, and the environment. However, the leading cause of most accidents comes from the risky behaviors of humans [39,43]. These behaviors can be as simple as speeding, drunk or drugged driving, cutting in too close in front, and dozing off while driving [44].

### 3.2. Data Collection

This study designs the research method for data gathering, consisting of a structured onsite survey (face-to-face). The respondents in this study had at least one year of experience driving or riding on the road within the study area which was evaluated, and the information gathered was based on the personal opinion of the respondents. The sample groups were divided into four categories based on the main modes of daily travel: private automobile (private car and motorcycle), active transportation (pedestrian and

bicycle), public transportation (public bus and mass transit (Bangkok Mass Transit System: BTS/Metropolitan Rapid Transit: MRT)), and paratransit (taxi and van) in the BMR. The study excluded people who had no experience in traveling in the study area. There were 3000 respondents who provided input for the analysis. Contributing factors to risk behavior perceptions in different travel modes provide the basis for the questionnaire survey design. The questionnaire consisted of four parts:

- Screen respondents to include only those who have driven or ridden on the roads within the study area within the last year.
- Obtain consent from the respondents.
- Construct road user profiles: socioeconomic characteristics (age, gender, education level, occupation, personal income level, and married status), attitude toward road safety, and driving experience (driving experience, possession of driving license, and accident experience).
- Assess their risk behavior perception: rule violation behaviors (illegal crossing, opposing driving lane usage, riding a motorcycle on a sidewalk, illegal U-turn, speeding, not slowing down in critical zones, illegal parking, running a red light, not using a seatbelt, overloading, driving after drinking alcohol, and not using a helmet), distraction behaviors (using a cell phone, listening to music, and smoking or eating while driving), fatigue (driving while taking drugs which can cause drowsiness), emotion (driving while aggressive or angry), and finally, personal errors (close following, not turning lights on, braking suddenly, driving so as to interfere with other vehicles).

As a final, critical step, Thammasat University, Thailand's Institutional Review Board (IRB), reviewed and approved this questionnaire (064/2022).

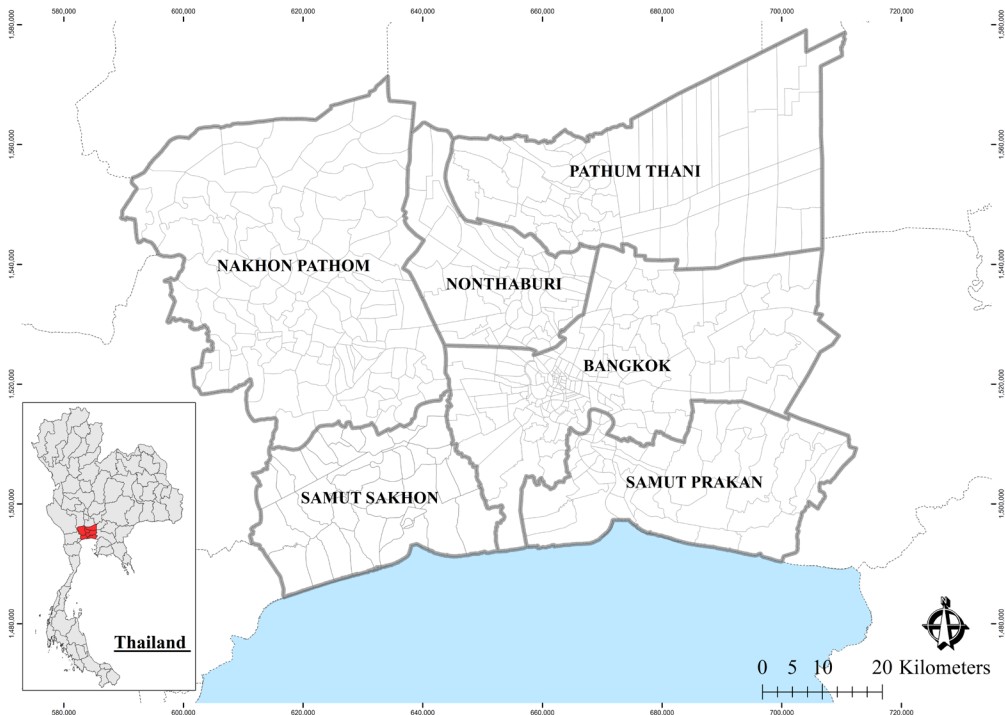

**Figure 1.** Study area.

### 3.3. Analysis

Prevailing safety interventions uncovered many aspects regarding the impacts of contributing factors to the risk of road traffic accidents on different travel modes with several contributing factors behind them. However, the association between them and realistic assessment of these factors present a few issues which must be investigated in-depth, especially in low- and middle-income countries. Thus, this study attempted

to examine the relationship of the perception of road users' risk behaviors in different travel modes (see Figure 2). Independent variables can be classified into three components: (1) socioeconomic characteristics (age, gender, education level, occupation, personal income, and marital status), types of these variables are categorical variables which can be nominal or ordinal; (2) attitude (attitude toward road safety), types of these variables are categorical variables as ordinal; and (3) driving experience (driving experience, possession of a driving license, accident experience), types of these variables present as categorical variables which can be ordinal or binary. For dependent variables, they can be represented as the score of the level of perception of risky behavior in which the type of variable is numeric (continuous).

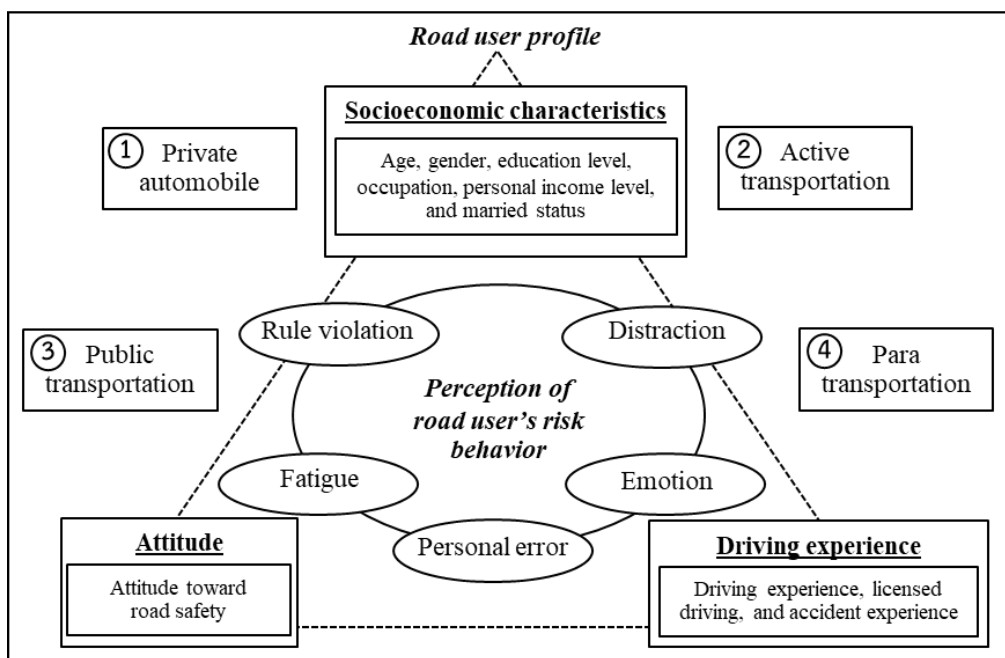

**Figure 2.** Framework.

Based on the central issue of road users' risk behaviors within different travel modes, this study designed the model for analysis by classifying it according to different travel modes. Therefore, the analysis comprises four general modes: (1) the personal vehicle (car and motorcycle), (2) active transportation (pedestrian and bicycle), (3) public transportation (bus and mass transit (sky train of BTS/subway of MRT)), and (4) paratransit (taxi and van). The statistical analyses were applied by using descriptive statistics and multiple regression analysis through the application of SPSS statistics (version 28.0).

The multiple regression analysis was employed to determine the relationships between two or more explanatory variables. This is because the performance of the model is an estimation based on the coefficient of determination (R-squared) value, where values closer to one indicate a better model [45]. Also, the F-test was used to test the statistical significance of the proposed model, and when the F-value presents as statistically significant ($p < 0.05$), it can be concluded that this model is a good fit with the data. Importantly, considering the regression coefficients, it is useful to represent the amount of the dependent variable changes in corresponding to changes in one unit of the independent variable. The regression coefficients can be negative or positive [46]. For a positive coefficient, the interpretation is that for every one-unit increase in the predictor variable, the outcome variable will increase by the coefficient value which contrasts with the negative coefficient. For each significance testing of the coefficient, a *p*-value can be used to determine at most equal to 0.05, if $p < 0.05$, it can be concluded that the coefficients are statistically significant to the prediction. Finally, variance inflation factor (VIF) was used to examine the collinearity

between independent variables. Generally, the study estimates VIF within the 1–5 score range. A value greater than five represents critical levels of multicollinearity [47–49].

## 4. Results

### 4.1. Socioeconomic Profile

Before considering and analyzing the data, understanding the characteristics of the respondents allows us to understand the context of the samples, which will differ from study to study. This study focused on the BMR megacity, a highly urbanized area that drives the country's economic activities. From the 3000 sets of respondents' travel experiences, all the sample groups had experience in road travel. The travel modes were divided into seven types, namely private car, motorcycle, pedestrian, bicycle, public bus, mass transit (BTS/MRT), and para transport (car taxi, motorcycle taxi, van, etc.) (See Tables 1 and 2).

Private vehicles, including private cars and motorcycles, represent the main travel mode of people living in the region, especially personal cars. The highest share of personal car users is young and middle-aged males who have a bachelor's degree. They represent the highest income level among all road users (541–675 USD). Men and women use motorcycles to an equal degree, similar to car users, and they are generally young and middle aged, half of them hold a bachelor's degree, and they have a moderately high personal income level (406–540 USD).

Walking, as one of the non-motorized active transportation modes, is slightly more popular among men than women. A bachelor's degree and vocational college are the most frequent education level attained among pedestrians having a moderately high personal income level (406–540 USD). In terms of cycling, the descriptive statistics showed that women are a few percentage points more active than men, but the difference is not significant. Concerning the education level, vocational college has the highest share, followed by a bachelor's degree. The lowest personal income level was found among the cyclists (271–405 USD).

**Table 1.** Respondents' social profiles.

| Variables | Travel Modes | | | | | | | | | | | | | |
| --- | --- | --- | --- | --- | --- | --- | --- | --- | --- | --- | --- | --- | --- | --- |
| | Private Automobile | | | | Active Transportation | | | | Public Transportation | | | | Paratransit | |
| | Private Car | | Motorcycle | | Pedestrian | | Bicycle | | Public Bus | | Mass Transit BTS/MRT | | | |
| | n | % | n | % | n | % | n | % | n | % | n | % | | |
| **Gender** | | | | | | | | | | | | | | |
| Male | 325 | 58.0 | 289 | 49.8 | 299 | 51.7 | 259 | 45.7 | 119 | 43.9 | 36 | 35.6 | 164 | 47.8 |
| Female | 200 | 35.7 | 253 | 43.6 | 249 | 43.1 | 274 | 48.3 | 114 | 42.1 | 54 | 53.5 | 143 | 41.7 |
| Others | 35 | 6.3 | 38 | 6.6 | 30 | 5.2 | 34 | 6.0 | 38 | 14.0 | 11 | 10.9 | 36 | 10.5 |
| **Age (year)** | | | | | | | | | | | | | | |
| 18–44 | 445 | 79.5 | 479 | 82.6 | 455 | 78.7 | 409 | 72.1 | 236 | 87.1 | 84 | 83.2 | 282 | 82.2 |
| 45–59 | 103 | 18.4 | 71 | 12.2 | 68 | 11.8 | 92 | 16.2 | 25 | 9.2 | 13 | 12.8 | 42 | 12.3 |
| 60 or over | 12 | 2.1 | 30 | 5.2 | 55 | 9.5 | 66 | 11.6 | 10 | 3.7 | 4 | 4.0 | 19 | 5.5 |
| **Marital status** | | | | | | | | | | | | | | |
| Married | 287 | 51.3 | 271 | 46.7 | 294 | 50.9 | 280 | 49.4 | 77 | 28.4 | 25 | 24.8 | 156 | 45.5 |
| Single | 217 | 38.7 | 256 | 44.2 | 223 | 38.6 | 235 | 41.5 | 145 | 53.5 | 57 | 56.4 | 138 | 40.2 |
| Divorced | 28 | 5.0 | 28 | 4.8 | 32 | 5.5 | 24 | 4.2 | 27 | 10.0 | 9 | 8.9 | 28 | 8.2 |
| Widowed | 28 | 5.0 | 25 | 4.3 | 29 | 5.0 | 28 | 4.9 | 22 | 8.1 | 10 | 9.9 | 21 | 6.1 |

Remark: 3000 sets.

**Table 2.** Respondents' economic profiles.

| Variables | Travel Modes | | | | | | | | | | | | | |
| --- | --- | --- | --- | --- | --- | --- | --- | --- | --- | --- | --- | --- | --- | --- |
| | Private Automobile | | | | Active Transportation | | | | Public Transportation | | | | Paratransit | |
| | Private Car | | Motorcycle | | Pedestrian | | Bicycle | | Public Bus | | Mass Transit BTS/MRT | | | |
| | n | % | n | % | n | % | n | % | n | % | n | % | n | % |
| Education level | | | | | | | | | | | | | | |
| Lower primary school | 0 | 0.0 | 0 | 0.0 | 0 | 0.0 | 1 | 0.2 | 0 | 0.0 | 0 | 0.0 | 0 | 0.0 |
| Primary school | 0 | 0.0 | 2 | 0.3 | 4 | 0.7 | 5 | 0.9 | 0 | 0.0 | 0 | 0.0 | 0 | 0.0 |
| Junior high school | 4 | 0.7 | 12 | 2.1 | 21 | 3.6 | 10 | 1.7 | 1 | 0.4 | 0 | 0.0 | 12 | 3.5 |
| High school | 65 | 11.6 | 91 | 15.7 | 121 | 20.9 | 97 | 17.1 | 54 | 19.9 | 20 | 19.8 | 61 | 17.8 |
| Vocational college | 112 | 20.0 | 204 | 35.2 | 211 | 36.5 | 250 | 44.1 | 74 | 27.3 | 23 | 22.8 | 123 | 35.9 |
| Bachelor's degree | 366 | 65.4 | 267 | 46.0 | 220 | 38.1 | 199 | 35.1 | 137 | 50.6 | 57 | 56.4 | 137 | 39.9 |
| Postgraduate | 13 | 2.3 | 4 | 0.7 | 1 | 0.2 | 5 | 0.9 | 5 | 1.8 | 1 | 1.0 | 10 | 2.9 |
| Income level (person/month) (USD) | | | | | | | | | | | | | | |
| Less than 135 | 5 | 0.9 | 11 | 1.9 | 20 | 3.5 | 22 | 3.9 | 14 | 5.2 | 2 | 2.0 | 7 | 2.1 |
| 135–270 | 25 | 4.5 | 44 | 7.6 | 58 | 10.0 | 74 | 13.0 | 23 | 8.5 | 7 | 6.9 | 32 | 9.3 |
| 271–405 | 88 | 15.7 | 139 | 24.0 | 158 | 27.3 | 188 | 33.2 | 83 | 30.6 | 30 | 29.7 | 90 | 26.2 |
| 406–540 | 162 | 28.9 | 209 | 36.0 | 167 | 28.9 | 101 | 17.8 | 45 | 16.6 | 20 | 19.8 | 77 | 22.4 |
| 541–675 | 196 | 35.0 | 103 | 17.7 | 112 | 19.4 | 110 | 19.4 | 47 | 17.3 | 17 | 16.8 | 79 | 23.1 |
| 676–810 | 34 | 6.1 | 38 | 6.6 | 20 | 3.5 | 26 | 4.6 | 28 | 10.3 | 10 | 9.9 | 26 | 7.6 |
| More than 810 | 50 | 8.9 | 36 | 6.2 | 43 | 7.4 | 46 | 8.1 | 31 | 11.5 | 15 | 14.9 | 32 | 9.3 |

Remark: 3000 sets; 1 United States dollar (USD) equates to 37.0439 baht (THB) in September, 2022, source from Bank of Thailand (2022).

Public transport comprises two modes: public bus and mass transit (BTS/MRT). Men and women use public busses to an equal degree, nevertheless mass transit is more popular among women. Young and middle aged are the most active groups within the widest age range, yet they contribute the highest percentage in public transportation modes. As with cyclists, public transport users have a lower personal income level than private car or motorcycle owners and pedestrians.

Among the elderly, 60 years of age and over, active modes such as walking and cycling are the most favored.

Based on socioeconomic characteristics, the data shows that gender differences are significant for modal travel usage. Men are more likely to travel in cars and motorcycles, while women use bicycles and public transportation more often, especially mass transit. However, when considering the income issue, the sample group found the highest income level of 406–540 USD, followed by 271–405 USD, and 541–675 USD, respectively. Different income levels reflect differences in people's abilities to access different modes of transportation. High-income respondents (541–675 USD and above) prefer to use their car, while lower-income respondents (406–540 USD and below) ride their motorcycle, bicycle, bus or BTS/MRT, or they walk. Therefore, in planning and developing road safety equality, it is necessary to balance the suitable choice of travel and the opportunity to access the chosen mode of travel for all groups of users, with good quality of service.

Before considering the results, the collinearity between independent variables was examined by the variance inflation factor (VIF). The variance inflation factor (VIF) was observed to lie within 1.040–2.427, indicating a low correlation among the variables, generally, as the study pinpoints the value within a range of scores from 1 to 5 [47–49]. Therefore, the input factors in the analysis were deemed appropriate to the analysis. For the private vehicle model, factors that were significant for perceived risk behaviors were gender (−0.118 *), education level (0.071 *), marital status (−0.208 **), attitude toward road safety

(−0.108 **), and accident experience (−0.213 **). Regarding active transportation, factors that were significant for perceived risk behaviors were age (−0.117 **), education level (0.053 *), occupation (−0.024 *), personal income level (−0.029 *), marital status (−0.284 **), driving experience (0.003 **), and possession of a driving license (−0.436 **). In terms of public transportation, factors that were significant for perceived risk behaviors were gender (−0.088 *), personal income level (0.047 *), marital status (−0.090 *), possession of driving license (−0.23 5*), and accident experience (−0.353 *). Finally, paratransit factors that were significant for perceived risk behavior were gender (−0.183 **), married status (−0.218 **), and possession of a driving license (−0.340 **).

### 4.2. Perception of Risk Behaviors and Travel Modes

Perception of risky behavior is an important aspect that reflects perspective and the importance of safe travel. Understanding road users' risky behaviors can lead to an accident-preventive approach to reduce both frequency and severity of accidents. In this study, the perception of the risky behaviors of different road users by different vehicles was considered together with their risky behavior. The perception of risky behavior is divided into five types:

1.　Rule violation;
2.　Distraction (using a cell phone, listening to music, driving while smoking or eating);
3.　Fatigue (due to taking drugs, driving when sleepy);
4.　Emotion (when angry or aggressive);
5.　Personal error (following too closely, no use of turn signal, interfering with other vehicles).

The score of the level of perception of risky behavior was determined based on six points, from zero to five. A zero score represents the perception that a situation will not cause any traffic accidents, while a score of five represents the perception that a situation causes serious accidents and may result in fatality and severe injury. Figure 3 demonstrates the different perceptions of risk behaviors and travel modes.

The survey found that road users perceived risk behaviors associated with fatigue in road accidents, with an average of 3.78, followed by emotion ($\bar{x}$ = 3.67), rule violation ($\bar{x}$ = 3.58), personal ($\bar{x}$ = 3.52), and distraction ($\bar{x}$ = 3.27). Road users with experience in driving vehicles and active transport travel perceived the riskiest behavior in travel. Most road users understand risky behavior at a moderate level, with an average of 3.56 (see Table 3). Their attitude toward road accidents is that there will be no fatalities, or that there may be only situations from such risky behaviors with serious injuries that require hospitalization. Risky behaviors related to fatigue include driving while taking drugs that cause drowsiness, and driving while feeling sleepy; they are considered violent risky behaviors because of the ability of the driver to lose control. These may lead to more death than do risky behaviors in a state where the driver is conscious of controlling the situation. However, sometimes decisions made when traveling may constitute driver's negligence, especially the violation of traffic rules.

**Table 3.** Overall data on the perception of risk behaviors and travel modes.

| Perception of Risk Behavior | AVG. by Travel Modes | | | | | | | Trend | Total AVG. | MAX | MIN |
|---|---|---|---|---|---|---|---|---|---|---|---|
| | A | M | P | BC | B | BM | PT | | | | |
| Rule violation | 3.58 | 3.57 | 3.65 | 3.70 | 3.39 | 3.33 | 3.53 | | 3.58 | 5 | 1 |
| Distraction | 3.19 | 3.17 | 3.32 | 3.50 | 3.12 | 3.29 | 3.22 | | 3.27 | 5 | 0 |
| Fatigue | 3.75 | 3.74 | 3.84 | 3.99 | 3.38 | 3.53 | 3.84 | | 3.78 | 5 | 1 |
| Emotion | 3.66 | 3.56 | 3.70 | 3.92 | 3.27 | 3.46 | 3.80 | | 3.67 | 5 | 0 |

**Table 3.** *Cont.*

| Perception of Risk Behavior | AVG. by Travel Modes | | | | | | | Trend | Total AVG. | MAX | MIN |
|---|---|---|---|---|---|---|---|---|---|---|---|
| | **A** | **M** | **P** | **BC** | **B** | **BM** | **PT** | | | | |
| Personal error | 3.59 | 3.48 | 3.57 | 3.61 | 3.33 | 3.03 | 3.47 | | 3.52 | 5 | 1 |
| Average | 3.55 | 3.50 | 3.62 | 3.74 | 3.30 | 3.33 | 3.57 | | 3.56 | 5 | 1 |

Remark: A = private vehicle, M = motorcycle, P = pedestrian, BC = bicycle, B = public bus, BM = mass transit (BTS/MRT), PT = paratransit.

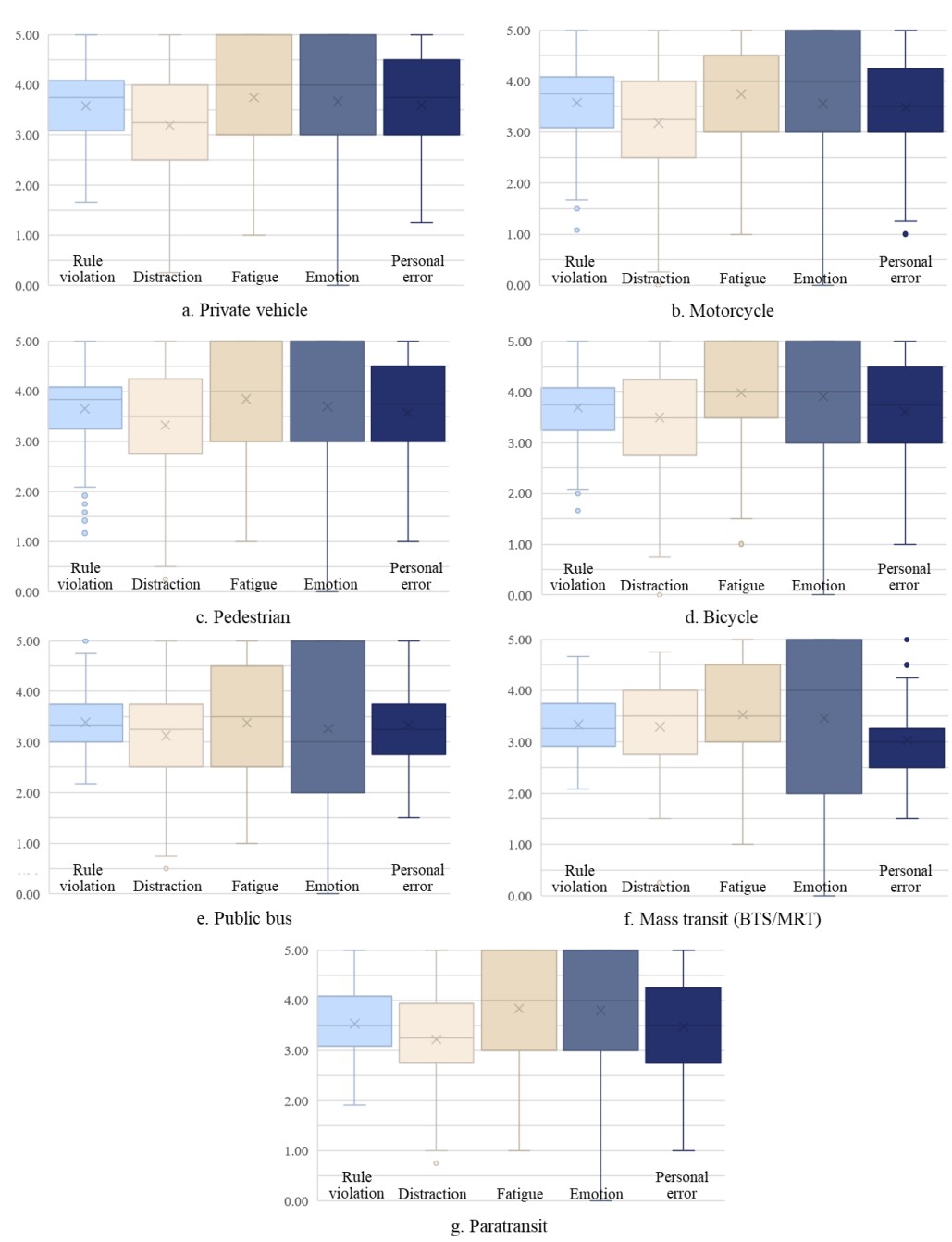

**Figure 3.** Perception of risk behaviors in different travel modes.

### 4.3. Risk Behavior on Differences in Travels Modes

The analysis divided responses into four general modes: private vehicles, active transportation, public transportation, and paratransit. The division allowed a closer evaluation

of relationships between the different modes of travel. Table 4 demonstrates the details of different models, and the result of the analysis.

**Table 4.** Factors affecting risk behavior on different travel modes.

| Variables | Model 1 | | | Model 2 | | | Model 3 | | | Model 4 | | |
|---|---|---|---|---|---|---|---|---|---|---|---|---|
| | Private Vehicle | | | Active Transportation | | | Public Transportation | | | Paratransit | | |
| | Coefficient | Std. Error | VIF | Coefficient | Std. Error | VIF | Coefficient | Std. Error | VIF | Coefficient | Std. Error | VIF |
| Socioeconomic characteristics | | | | | | | | | | | | |
| Age | 0.027 | 0.041 | 1.178 | −0.117 ** | 0.030 | 1.167 | −0.024 | 0.054 | 1.078 | 0.017 | 0.052 | 1.153 |
| Gender | −0.118 * | 0.036 | 1.069 | −0.024 | 0.031 | 1.049 | −0.088 * | 0.045 | 1.104 | −0.183 ** | 0.046 | 1.156 |
| Education level | 0.071 * | 0.029 | 1.167 | 0.053 * | 0.022 | 1.137 | −0.040 | 0.038 | 1.140 | 0.002 | 0.033 | 1.077 |
| Occupation | −0.007 | 0.016 | 1.073 | −0.024 * | 0.013 | 1.186 | −0.023 | 0.022 | 1.099 | −0.001 | 0.019 | 1.072 |
| Personal income | −0.012 | 0.018 | 1.191 | −0.029 * | 0.014 | 1.289 | 0.047* | 0.024 | 1.809 | −0.020 | 0.022 | 1.262 |
| Marital status | −0.208 ** | 0.029 | 1.117 | −0.284 ** | 0.024 | 1.084 | −0.090* | 0.038 | 1.187 | −0.218 ** | 0.036 | 1.181 |
| Attitude | | | | | | | | | | | | |
| Attitude toward road safety | −0.108 ** | 0.028 | 1.066 | −0.045 | 0.025 | 1.040 | −0.040 | 0.049 | 1.105 | −0.146 * | 0.042 | 1.140 |
| Driving experience | | | | | | | | | | | | |
| Driving experience | −0.002 | 0.003 | 1.268 | 0.015 ** | 0.003 | 1.654 | 0.002 | 0.009 | 2.427 | 0.003 | 0.005 | 1.872 |
| Possession of a driving license | −0.090 | 0.129 | 1.084 | −0.436 ** | 0.050 | 1.838 | −0.235 * | 0.081 | 1.671 | −0.340 ** | 0.084 | 2.163 |
| Accident experience | −0.213 ** | 0.045 | 1.093 | −0.051 | 0.041 | 1.148 | −0.353 * | 0.083 | 1.776 | −0.060 | 0.075 | 1.348 |
| R Square | 0.33 | | | 0.47 | | | 0.16 | | | 0.31 | | |
| F-ratio | 13.97, $p < 0.001$ | | | 32.13, $p < 0.001$ | | | 6.94, $p < 0.001$ | | | 15.06, $p < 0.001$ | | |

Remark: * = significant level at 0.05; ** = significant level at 0.001; Df = 10.

Before considering the results, the collinearity between independent variables was examined by the variance inflation factor (VIF). The variance inflation factor (VIF) was observed to lie within 1.040–2.427, indicating a low correlation among the variables, generally, as the study pinpoints the value within the range of 1–5 [47–49]. Therefore, the input factors in the analysis were deemed appropriate in the analysis. For the private vehicle model, factors that were significant for perceived risk behavior were gender (−0.118 *), education level (0.071 *), marital status (−0.208 **), attitude toward road safety (−0.108 **), and accident experience (−0.213 **). Regarding active transportation, factors that were significant for perceived risk behavior were age (−0.117 **), education level (0.053 *), occupation (−0.024 *), personal income level (−0.029 *), marital status (−0.284 **), driving experience (0.003 **), and possession of driving license (−0.436 **). In terms of public transportation, factors that were significant for perceived risk behavior were gender (−0.088 *), personal income level (0.047 *), marital status (−0.090 *), possession of driving license (−0.235 *), and accident experience (−0.353 *). Finally, paratransit factors that were significant for perceived risk behavior were gender (−0.183 **), married status (−0.218 **), and possession of a driving license (−0.340 **).

## 5. Discussion

Vehicle vulnerability concerns road safety planners regarding the severity of accidents. Different vehicles have different levels of physical protection, while some vehicles used for road travel do not have protective barriers, such as motorcycles, bicycles, or pedestrians. Therefore, identifying the driving factor that causes users of roads to perform risky, intentional, and unintentional behavior, is essential. According to the study of Rella

Riccardi et al. [50], drivers' behaviours and psychophysical states turned out to be crucial patterns related to the overrepresentation of pedestrian crashes. These factors are all related to human factors, in which each person is fundamentally different in their demographic, social, and economic characteristics. Other factors are related to the environment, such as the physical environment that drives risky behaviors or the impact of social and family influences on personal attitudes and views toward road safety. However, road safety studies comparing travel patterns have begun to appear in the policy literature, for example, comparing risks posed by vans and lorries [51], and comparing risks in different modes of transport [11].

Studying the differences in travel patterns is not a new issue, but the complexity of contexts related to different personal factors in each context remains to be comprehensively understood. Although it is widely studied abroad, this issue is still less common in Thailand in road safety studies.

The survey revealed only minor differences among the average values of the perceived importance of various risky behaviors. This is a logical consequence of the quite complex task given to the respondents. It is really not easy to decide, whether fatigue or personal error is riskier. In addition, even in cases of in-depth accident analyses, it is also difficult to identify the type of risky behavior, as in many cases more than one type of behavior played a role. Considering the ranges of the perceived risks, interesting differences can be observed. The assessment of the violation of rules is the most homogeneous among respondents, while in the perception of emotion there are much larger differences. Another point is that risky behaviors are culture-dependent. Both the frequency of these behaviors and their perceived importance might be different among different countries. Further research can reveal such differences. Contributing factors regarding road users' characteristics include: age, gender, education level, personal income level, marital status, driving experience, accident experience, and attitude toward road safety, all of which affect risk behavior perceptions over different travel modes. However, when considering factors that increase awareness of risky behavior in travel, it was found that the factors that significantly increase the level of perception were education level, driving experience, and income level. This study concords with the study of Crundall [52], pinpointed to experience drivers' relationships with hazard perception or the ability to predict dangerous situations on the road [53]. Alonso et al. [54] found that risky behavior can be changed according to the sociodemographic characteristic, including attitude towards road safety. Borowsky et al. [55] also revealed that driving experience is correlated with risk perception. When considering the issue of travel mode type, it was found that travel modes have significant effects on driving risk perception. The findings in our study concord with the study of Sahebi et al. [56], who studied the factors affecting driving risk perception, and found that vehicle type, driving experience, and education level each have significance relationships with driving risk perception. Furian et al. [57,58] estimated the contributing factors to road accidents and perceived risky behaviours of other road users, which pinpointed that drivers are aware of risky behaviors among those who travel on various modes of travel. The majority of risky behaviors that travelers perceive to be at risk involve behaviors caused by human error, for example: driving under the influence of alcohol, driving under the influence of drugs, or tiredness This issue is consistent with the perceptions of risk behaviors of the respondents in this study as well. Notably, focusing on increasing awareness and perceptions of risky behaviors in travel is an important issue, which in many studies indicated that low awareness and risk perception is associated with an increased crash risk [59,60]. Additionally, the results of the study suggested the importance and awareness of the risks of people in the group of active transport, which is perceived more than any other mode of transport. It may be that these travelers are aware of their own dangers and vulnerabilities in the event of an accident. This is an interesting point because the mode of active transport is becoming extremely important nowadays. Although people turn to travel with active transport more, and these groups are aware of travel safety, they are still faced with the challenges of road safety, especially when traveling on a road with other modes [61]. Thus, urban and transport

planning should be focused on the dimensions of convenience, accessibility, and road safety, for all users and all modes of transport [62]. The behavior perceptions mentioned above relate to individual personality or human factors, often expressed as human errors and playing a significant role in traffic accidents [63]. However, the planning for solving problems should also have a comprehensive plan for all risks in other dimensions. This study also provides information in different contexts of individual characteristics and geographical contexts; however, the results are consistent with several studies suggesting the relevance of the contributing factors to road safety issues in different modes of travel [64]. With comprehensive understanding among these contributing factors, advocacy for road safety policies could be effectively utilized for improving the achievement of the sustainable development goals by targeting more inclusive, high quality transport infrastructure, with sustainable road fatality reduction.

## 6. Conclusions

This study focused on road users' perceptions of risk behaviors within different travel modes. Examining the risk behaviors of commuters in the Bangkok Metropolitan Region contributed to understanding the factors affecting the perceptions of different risk behaviors among travel modes in urban areas and middle-income countries concerned with accident death rates. By studying data from 3000 questionnaires, contributing factors to risk behavior perception in different travel modes correlated to the following road user characteristics: socioeconomic, attitude toward road safety, driving experience, and risk behavior perception (rule violation behavior, distraction, fatigue, personal error). The critical aspects of this study point out that road users in vulnerable modes of travel, such as walking and cycling, were more aware of risky behaviors than those in other modes of travel. However, all road users understood risky behavior at a moderate level.

Moreover, road users do not envision fatalities or other severe situations from such risky behaviors. Therefore, it is imperative to foster a safe culture so that road users know the actual risks of travel that can lead to death and severe injury. Further research should consider the differing attitudes of road users and the establishment of a travel-safety culture. Different travel modes contribute to the fundamental factors driving different behaviors, attitudes, and perceptions. That is to say, both internal and external norm factors play a role in supporting behaviors that lead to risky driving.

**Author Contributions:** Conceptualization, P.I.; Methodology, P.I. and E.M.; Formal analysis, P.I., S.C. and E.M.; Investigation, S.C.; Data curation, S.C.; Writing—original draft, P.I., S.C., E.M. and S.P.; Writing—review & editing, P.I., S.C., E.M. and S.P.; Visualization, S.C.; Supervision, P.I. and E.M.; Funding acquisition, P.I. All authors have read and agreed to the published version of the manuscript.

**Funding:** This research was funded by Thammasat University Research Fund under the Thailand Science Research and Innovation (TSRI) with Fundamental Fund (2022) and also partially supported by Thammasat University Research Fund with Fast Track (2022), Contract No. TUFT-FF23/2565). The APC was funded by Széchenyi István University.

**Institutional Review Board Statement:** The study was conducted according to the guidelines of the Declaration of Helsinki and approved from the Human Research Ethics Committee of Thammasat University Social Sciences (certificate of approval number 064/2022, 18 July 2022).

**Informed Consent Statement:** Informed consent was obtained from all subjects involved in the study.

**Data Availability Statement:** Not applicable.

**Acknowledgments:** The authors gratefully acknowledge the support provided by Thammasat University Research Fund under the Thailand Science Research and Innovation (TSRI) with Fundamental Fund (2022) of a project entitled "The Holistic Approach of Urban Planning toward An Integrative Urban Land Use and Road Safety", Contract No. TUFF 53/2565). It is also partially supported by Thammasat Uni-versity Research Fund with Fast Track (2022), Contract No. TUFT-FF23/2565), and conducted by Center of Excellence in Urban Mobility Research and Innovation, Faculty of Architecture and Plan-ning, Thammasat University, Pathumthani, Thailand.

**Conflicts of Interest:** The authors declare no conflict of interest.

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
