# Peer review of "Analysis of Road Users’ Risk Behaviors in Different Travel Modes: The Bangkok Metropolitan Region, Thailand"

_infrastructures, doi:10.3390/infrastructures8040079_

Round 1
Reviewer 1 Report
The topic of this paper is related to the theme of the journal, and the manuscript is well structured and is worth publishing.
The topic is original and relevant. This study analyzes the risk behaviors of road users in different modes of transportation in the Bangkok metropolitan area.
It also provides us with information about the relevant factors affecting risk behaviors in different modes of transportation, but it does not give us solutions to the safety problems.
The study is scientifically sound and, in my humble opinion, is aimed at researchers new to the field of road traffic safety. The references are adequate, and the conclusions should include information about who this study is aimed at and some suggestions for further research.
The manuscript needs some minor changes in text, and here are the suggestions:
Paragraph 97 – rephrase the sentence (…the causes of the casualty were multiple causes…)
Paragraph 134 – please explain what the difference is between low and middle-income countries (GDP range).
Paragraph 137-139 – please write down how many traffic accidents and fatalities occurred between 2017 and 2021 for all travel modes.
Table 1 and 2 are not referenced in the text.
Table 1 The sum of n is 2999, private car and gender sum of percentage is 102.1 (58+35,8+8.3), and the sum of marital status is 99.9.
Paragraph 208 – 209 and other – please write how much is personal income 10000, 15000, 20000 or 25000 baht in US$ or in EU€.
Section 4.3. Please comment WIF values from Table 4 in text.
Paragraph 325 – rephrase sentence (Regarding considering…), to make it more understandable.
Paragraph 332 – rephrase sentence (This study provides…), to make it more understandable.
Section 6. Conclusions should include information about who this study is aimed at and some suggestions for further research.
Paragraph 337 - rephrase sentence (This study explores road users' behavior and road accident analysis …) this study was not about road accident analysis.
Reviewer 2 Report
The authors analysed the risk behaviour of road users in different travel modes in Bangkok. The study is interesting, the data well organized, and the results are reasonable.
Below, I provided some comments with the aim to improve the paper.
- Please replace reduce with halve in the sentence at lines 49-50 and introduce that the target is to halve the fatalities and the serious injuries as well.
- Please, clearly state the aims of your research at the end of the introduction section.
- Data collection section. How was the onsite survey widespread? How did you evaluate if a person had experience in traveling in the study area? Was it based just on the personal opinion of the people? How were the questions structured?
- Analysis. How did you perform you models? Please, state it in the paragraph. Furthermore, introduce the p-values to understand the significance of the results and the odds ratios to understand the impact. The odds ratios are explained here https://doi.org/10.3390/su142215471, https://doi.org/10.1016/j.aap.2020.105782, https://doi.org/10.1016/j.jsr.2008.06.003 for inspiration).
- in the results, please, add the Rsquare of Mc Fadden and the total number of observations in table 4. For each variable, please also state the baseline.
- The discussion should be expanded in their comparison with the existing literature and the main results of this study should be emphasized. For instance, the active transport is becoming extremely important nowadays. In the 2030 agenda for sustainable development, indeed, it was said that the cities and human settlements need to be make inclusive, safe, resilient and sustainable for all users and all modes of transport. Hence, substantial changes in the way the roads are designed in crucial to prevent crashes with vulnerable road users (https://doi.org/10.3390/su142013142, https://doi.org/10.1016/j.aap.2023.106996, https://doi.org/10.1016/j.iatssr.2020.08.006)
Minor
The structure of some sentences should be revised. For instance:
- Lines 37-39, “the complexity of the transportation system, not only from the perspective of transportation dimensions and sizes but also the complexity of activities 38 within the environment in the transport routes”.
- Lines 87-88, “Driver behavior, risk behavior, is a crucial cause of accidents, with past studies suggesting that this is a contributory factor of road traffic accidents among all road users”.
- Lines 265-267, “a score of 5 means the perception that such situations will cause to cause serious accidents and may result in death and severe injury”.
- Line 325, “Regarding considering different travel modes”.
Add references were needed. For instance:
- Lines 80-81, “Casualty from road traffic accidents is the leading cause of death for road users in all 80 age groups. More than half of all are vulnerable road users”.
Please, introduce the acronym before using it. For instance:
- Lines 200, BTS/MRT.
Please, update the number of the chapters and their paragraphs, namely the sections Data collection and Analysis.
Round 2
Reviewer 2 Report
The authors' answers to my concerns and issues are not exhaustive. The paper lacks a useful discussion section. Furthermore, in their responses they just provided the lines were the answers were provided but they do not match.
In addition, several comments have not been addressed at all.
- How did you evaluate if a person had experience in traveling in the study area? Was it based just on the personal opinion of the people?
- Analysis. State the software you use to perform you models.
- Introduce the p-values to understand the significance of the results and the odds ratios to understand the impact (https://doi.org/10.3390/su142215471, https://doi.org/10.1016/j.aap.2020.105782).
- For each variable in Table 4, please state the baseline. how did you consider the varables in your study? nominal or numeric?
- 1. The discussion should be expanded in their comparison with the existing literature and the main results of this study should be emphasized. For instance, the active transport is becoming extremely important nowadays. In the 2030 agenda for sustainable development, indeed, it was said that the cities and human settlements need to be make inclusive, safe, resilient and sustainable for all users and all modes of transport. Hence, substantial changes in the way the roads are designed in crucial to prevent crashes with vulnerable road users (https://doi.org/10.3390/su142013142, https://doi.org/10.1016/j.aap.2023.106996, https://doi.org/10.1016/j.iatssr.2020.08.006)
Author Response
Please find the enclosed document

Round 3
Reviewer 2 Report
Thank you for your contribution. The paper has improved and I found it eligible for publication. Good luck!